# Monkeypox Knowledge and Confidence in Diagnosis and Management with Evaluation of Emerging Virus Infection Conspiracies among Health Professionals in Kuwait

**DOI:** 10.3390/pathogens11090994

**Published:** 2022-08-31

**Authors:** Mariam Alsanafi, Kholoud Al-Mahzoum, Malik Sallam

**Affiliations:** 1Department of Pharmacy Practice, Faculty of Pharmacy, Kuwait University, Kuwait City 25210, Kuwait; 2Department of Pharmaceutical Sciences, Public Authority for Applied Education and Training, College of Health Sciences, Safat 13092, Kuwait; 3School of Medicine, The University of Jordan, Amman 11942, Jordan; 4Department of Pathology, Microbiology and Forensic Medicine, School of Medicine, The University of Jordan, Amman 11942, Jordan; 5Department of Clinical Laboratories and Forensic Medicine, Jordan University Hospital, Amman 11942, Jordan; 6Department of Translational Medicine, Faculty of Medicine, Lund University, 22184 Malmö, Sweden

**Keywords:** public health emergency of international concern, epidemic, preparedness, response

## Abstract

As the 2022 human monkeypox (HMPX) multi-country outbreak is spreading, the response of healthcare workers (HCWs) is central to mitigation efforts. The current study aimed to evaluate HMPX knowledge and confidence in diagnosis and management among HCWs in Kuwait. We used a self-administered questionnaire distributed in July–August 2022 through a snowball sampling approach. The survey items evaluated HMPX knowledge, confidence in diagnosis and management of the disease, and the belief in conspiracies regarding emerging virus infections (EVIs). The sample size was 896 HCWs: nurses (*n* = 485, 54.1%), pharmacists (*n* = 154, 17.2%), physicians (*n* = 108, 12.1%), medical technicians/allied health professionals (MT/AHP, *n* = 96, 10.7%), and dentists (*n* = 53, 5.9%). An overall low level of HMPX knowledge was noticed for items assessing virus transmission and non-cutaneous symptoms of the disease, with higher knowledge among physicians. Approximately one-fifth of the study sample agreed with the false notion that HMPX is exclusive to male homosexuals (*n* = 183, 20.4%), which was associated with lower knowledge with higher frequency among MT/AHP compared to nurses, physicians, and pharmacists. Confidence levels were low: confidence in diagnosis based on diagnostic tests (*n* = 449, 50.1%), confidence in the ability to manage the HMPX (*n* = 426, 47.5%), and confidence in the ability to diagnose HMPX clinically (*n* = 289, 32.3%). Higher confidence levels were found among nurses and participants with postgraduate degrees. Higher embrace of conspiracy beliefs regarding EVIs was noticed among participants with lower knowledge, and among those who agreed or were neutral/had no opinion regarding the false idea of HMPX exclusive occurrence among male homosexuals, while lower levels of belief in conspiracies were noticed among physicians, dentists, and pharmacists compared to MT/AHP. Variable levels of HMPX knowledge were observed in this study per item, with low level of knowledge regarding virus transmission. Differences in knowledge and confidence levels in diagnosis and management of HMPX should be considered in education and training aiming to prepare for outbreak response. The relatively high prevalence of embracing conspiratorial beliefs regarding EVIs is worrisome and needs proper interventions. The attitude towards male homosexuals’ role in monkeypox spread should be evaluated in future studies considering the possibility of stigma and discrimination in this most-at-risk group.

## 1. Introduction

The monkeypox virus (MPXV) has been reported to cause human monkeypox (HMPX) zoonotic infection since early 1970s [1,2]. The disease has been endemic in Central and Western Africa since then, with a few outbreaks that were reported in the US and the UK [3]. These outbreaks with limited spread were linked to imported animals from the endemic regions or travel history to these regions [4,5]. However, the latest 2022 HMPX multi-country outbreak has a different scenario, with more than 31,000 cases in 82 previously non-endemic countries and territories, in addition to 375 HMPX reported in seven endemic countries as of 11 August 2022 [6]. The rapid rate of disease spread and the need for a collaborative and well-coordinated response necessitated the declaration of the 2022 HMPX outbreak as a public health emergency of international concern [7,8].

The transmission of the MPXV occurs directly through close contact and indirectly via fomites [9,10,11,12,13]. Among the peculiarities of the ongoing HMPX outbreak is the observation of transmission clusters among men who have sex with men (MSM, including gays and bisexual men) [12,14,15,16,17]. This suggests that sexual transmission could be an effective mode of virus spread [18,19,20].

Clinically, HMPX is primarily a cutaneous disease with lymphadenopathy similar to the first infectious disease to be eradicated from humans; namely, smallpox [2,21]. The incubation period ranges between 5 and 21 days, followed by flu-like symptoms and the development of skin rash [21]. This skin eruption evolves through the following stages: macules, papules, pustules, vesicles, and scabs [22]. During the current HMPX outbreak, genital lesions have been frequently reported, with possibility of asymptomatic infection [16,22,23]. Hospital admission could be indicated considering the reporting of HMPX complications that include: super-infection by bacteria, dehydration, and respiratory distress, among others [12,24,25]. The case–fatality ratio has been reported up to 11%; however, only twelve mortalities have been linked to the disease amid the ongoing 2022 HMPX outbreak [2,3,6,10].

Antivirals are available for treatment, and prevention relies on vaccination [10,21,26,27]. In addition, the preventive efforts can benefit from the central role of healthcare workers (HCWs) through active surveillance and refined diagnosis and management, which can help in disease control [28]. Therefore, the assessment of HCWs’ knowledge and their current confidence levels to diagnose and manage HMPX can be helpful in guiding the response plans needed for the control and mitigation efforts [29].

The few previous and recent studies among HCWs, university students, and the general population that evaluated HMPX knowledge revealed defects in HMPX knowledge regarding the different aspects of the disease besides low levels of confidence to diagnose and manage the cases [30,31,32,33,34]. These findings can be understandable due to the previous lack of attention to the disease outside the endemic regions, lack of educational material about the topic in health schools’ curricula, and courses besides the lack of clinical training [31,32,33,34,35]. Further research is needed, particularly among health professionals based on their central role in outbreak response, and due to the rapid dissemination of the HMPX outbreak [36].

Recently, the frequent emergence of infectious diseases was accompanied by infodemics characterized by viral dissemination of misinformation, social media panic, and bizarre conspiracy ideas that could spread faster than the disease, itself [37,38]. The belief in conspiracies was conspicuous during the recent Ebola outbreaks and the coronavirus disease 2019 (COVID-19) pandemic [39,40,41,42]. Thus, the almost immediate circulation of conspiracies surrounding the 2022 HMPX outbreak could be considered an expected phenomenon [43,44,45]. Although the general embrace of conspiracy ideas could appear harmless, its potential harmful impact has been reported, particularly in the context of health-seeking behavior manifested in vaccination hesitancy and distrust in science and health institutions [42]. Therefore, the assessment of conspiracy belief pervasiveness, especially among HCWs, appears essential, considering their role in curtailing the negative influence of these harmful beliefs [46]. 

Kuwait is an Arab Middle Eastern high-income country with a population of more than 4,200,000 people. Although HMPX has not been reported in the country so far, the rapid increase of cases worldwide, besides the reporting of HMPX in other Arab countries (United Arab Emirates, Qatar, Lebanon, and Morocco) necessitates vigilant preparedness and response plans [47]. Considering the primary role of health professionals in responding to the ongoing HMPX outbreak [7,29,48], we aimed to assess the current knowledge of HCWs in Kuwait about the disease. In addition, the study goals included the evaluation of the current levels of confidence to diagnose and manage HMPX. Finally, we aimed to evaluate the attitude of HCWs in Kuwait towards conspiracies that are related to virus emergence and the subsequent measures aimed at controlling emerging viral diseases.

## 2. Materials and Methods

### 2.1. Study Design

This was a cross-sectional, web-based survey of HCWs in Kuwait. The target population included nurses, physicians, dentists, pharmacists, medical technicians (MTs), and HCWs in other allied health professions (AHPs), aged 18 years or older. The approval of the study was granted by the Health Sciences Center Ethical Committee at Kuwait University (reference number: VDR/EC-4063-1711). An electronic informed consent was obtained from all study participants, which was mandatory for completion of the survey.

The questionnaire was prepared in Arabic and English simultaneously and it was distributed via Google forms during 18 July 2022–9 August 2022. Convenience sampling started by the contacts of the first and second authors with requests to their contacts to share the survey link (snowball sampling). The following social media and instant messaging applications were used to promote participation in the study: Facebook, Twitter, Instagram, WhatsApp, and Messenger. Participation was voluntary, without incentives for participation or paid advertisement.

The minimum sample size was 595 participants based on the previous estimates on the number of HCWs in Kuwait and using the CheckMarket sample size calculator [49,50].

### 2.2. Survey Instrument

The survey comprised five sections, with a total of 32 items including the informed e-consent item: first, an introductory section with information about the study and its objectives followed by the e-consent for participation. Second, a section assessing the sociodemographic characteristics (age, sex, highest educational level (undergraduate degree vs. postgraduate degree), monthly income of the household in Kuwaiti dinar (KWD, divided into two categories: ≤1250 KWD or >1250 KWD), and occupational category (nurses, physicians, dentists, pharmacists, or MT/AHP)).

Third, a ten-item HMPX knowledge section was included, with items adopted from Harapan et al. assessing knowledge about the epidemiology, clinical aspects, transmission, and treatment of the disease [31]. Knowledge score (K-score) was calculated based on the sum score of the ten knowledge items with correct responses scored as (+1), I do not know scored as zero, while incorrect responses were scored as (−1), similar to the previous approach used in a study that investigated HMPX knowledge among university students in health schools in Jordan [34].

Fourth, a three-item section on the confidence of HMPX diagnosis and management was included with items that were adopted from Harapan et al. to assess confidence levels to diagnose and manage HMPX based on the current level of knowledge and skills and to diagnose the disease based on diagnostic tests [32]. A confidence score was calculated based on the sum of the three responses from each respondent, with “yes” scored as (+1) while “no” was scored as zero [51].

Finally, a twelve-item section, which assessed the conspiratorial attitude towards emerging virus infections and the subsequent intervention measures beside an item assessing the attitude to the false idea that HMPX occurs exclusively among male homosexuals, was included. The emerging virus infections conspiracy scale (EVICS) used to assess the embrace of these conspiratorial ideas towards virus emergence was adopted from Freeman et al. and it was previously used in Arabic language by Sallam et al. in the context of HMPX outbreak [34,52].

The EVICS items were scored from 1 to 7 based on its measurement on a 7-point Likert scale: 1 = strongly disagree, 2 = disagree, 3 = somewhat disagree, 4 = neutral/no opinion, 5 = somewhat agree, 6 = agree, and 7 = strongly agree. Accordingly, higher EVICS indicated a higher embrace of conspiracies regarding emerging virus infections [34,51]. Response to all items in the survey was mandated for successful submission of the completed questionnaire. This approach was done to eliminate the issue of item non-response bias. 

### 2.3. Statistical Analysis

All statistical analyses were conducted using IBM SPSS Statistics for Windows (Version 22.0. Armonk, NY: IBM Corp). Descriptive statistics included the measurements of mean and standard deviation (SD). Univariate analyses were conducted based on the Chi-squared (χ^2^), Mann–Whitney *U* (M-W), and Kruskal–Wallis (K-W) tests. Logistic regression multivariate analyses were done to evaluate the associations between different study variables as follows: The K-score was dichotomized based on the mean value of the entire study sample into two categories; K-score < 4 (inferior HMPX knowledge) vs. K-score ≥ 4 (better HMPX knowledge). The attitude towards the false belief that HMPX is exclusive among male homosexuals was divided into two categories: participants who strongly disagreed, disagreed or somewhat disagreed with the attitude statement (disagreement) vs. those who strongly agreed, agreed, or somewhat agreed with the attitude statement or were neutral/had no opinion (agreement/neutral/no opinion). The confidence score was dichotomized into those with a score of zero or 1 (indicating lower confidence to diagnose and manage HMPX) vs. those with a score of 2 or 3 (indicating higher confidence to diagnose and manage HMPX). The EVICS score was divided into two categories based on the mean value in the whole study sample: An EVICS ≥ 46 indicated a higher embrace of conspiracies regarding emerging virus infections vs. EVICS < 46 indicating a lower embrace of conspiracies. The statistical significance was considered for *p* < 0.050. 

## 3. Results

### 3.1. Description of the Study Sample

The final study sample had 896 HCWs with a majority of females (*n* = 650, 72.5%) and nurses (*n* = 485, 54.1%) with a mean age of 35.4 years (SD = 7.4).

Characteristics of the study sample divided by occupational category are shown in (Figure 1). The unequal distribution of the study participants regarding age, sex, educational level and monthly income was observed upon comparison based on occupational category (*p* < 0.001 for all comparisons, Figure 1).

### 3.2. Higher Level of HMPX Knowledge Was Found among Physicians Compared to Other HCWs

For the entire study sample, the mean K-score was 3.8 (SD = 2.5). Statistically significant differences were observed based on occupational category, with physicians having the highest K-score (mean = 4.6 ± 2.3), followed by pharmacists (mean = 4.2 ± 2.3), dentists (mean = 4.1 ± 2.3), nurses (mean = 3.5 ± 2.6), and MT/AHP (mean = 3.5 ± 2.3, *p* < 0.001, K-W). Participants with a monthly income > 1250 KWD had a higher mean K-score (4.1) compared to those with a monthly income ≤ 1250 KWD (mean K-score = 3.6, *p* = 0.012, M-W). No statistically significant differences were observed in K-score based on age (*p* = 0.884, M-W), sex (*p* = 0.076, M-W), or educational level (*p* = 0.126, M-W).

Despite the variability in the percentage of correct responses per item, three knowledge items were the top three in correct response among all occupations, as follows: “monkeypox is caused by a virus”; “papules, vesicles and pustules on the skin are signs of human monkeypox”; and “there are many human monkeypox cases in Kuwait” (incorrect, Figure 2). On the other hand, two items were in the bottom two items with correct responses among the majority of occupational categories: “human-to-human transmission of monkeypox occurs easily” (incorrect); and “diarrhea is one of the signs or symptoms of human monkeypox” (incorrect).

Multinomial logistic regression analysis showed that occupation was the only significant factor associated with better HMPX knowledge, which was reported among physicians compared to MT/AHP as the reference group (Table 1).

### 3.3. Attitude towards the False Notion That HMPX Is Exclusive to Male Homosexuals

Approximately half of the study sample either disagreed strongly, disagreed, or somewhat disagreed that HMPX is exclusive to male homosexuals (*n* = 442, 49.3%), while less than a third were neutral or had no opinion (*n* = 271, 30.2%). Differences in attitude to this item were observed based on occupational category (*p* < 0.001, χ^2^ = 86.326, Figure 3).

Logistic regression analysis showed that occupation and HMPX knowledge were the only factors that were significantly associated with disagreement regarding the idea of HMPX exclusivity among male homosexuals (Table 2).

### 3.4. Low Confidence of HCWs in Kuwait to Diagnose and Manage HMPX

Approximately half of the study sample were confident to diagnose HMPX based on the ability of their institutions to conduct the diagnostic tests for MPXV (*n* = 449, 50.1%). A closer confidence level was observed for the self-reported ability to manage the HMPX cases based on their current levels of skills and knowledge (*n* = 426, 47.5%). However, less than a third of the study sample were confident in their ability to diagnose HMPX based on their current knowledge and skills (*n* = 289, 32.3%). Responses to the three items that assessed confidence to diagnose and manage HMPX divided by occupational categories is shown in (Table 3).

Using the confidence score, significantly higher confidence levels to diagnose and manage HMPX were observed among nurses, while dentists showed lower confidence levels with MT/AHP as the reference group (Table 4). In addition, participants with postgraduate educational level showed significantly higher confidence scores (Table 4).

### 3.5. Attitude towards Conspiratorial Beliefs Regarding Emergence of Virus Infections

The mean EVICS score of the entire study sample was 45.35, with variability in attitude-per-item as follows: the highest mean was found for the items “I am skeptical about the official explanation regarding the cause of virus emergence,” mean = 4.78 and “Viruses are biological weapons manufactured by the superpowers to take global control,” mean = 4.16, while the lowest mean was found for the item “Coronavirus was a plot by globalists to destroy religion by banning gatherings,” mean = 3.28. The variability in responses per item in the whole study sample is shown in (Figure 4), while the responses to the twelve EVICS items stratified per occupational category is shown in (Table 5).

Multinomial regression analysis showed a higher embrace of conspiracy beliefs regarding emerging virus infections among females, participants with lower HMPX knowledge, and those who agreed or had no opinion regarding the exclusivity of HMPX occurrence among male homosexuals (Table 6).

## 4. Discussion

The current study revealed the presence of knowledge gaps among HCWs in Kuwait regarding the HMPX infection. This result was found despite ubiquitous media coverage tackling the topic, as well as the rapid and timely delivery of published literature addressing almost every single aspect of the disease [9,12,14,15,17,19,21,22,53]. Therefore, the design of efficient and well-organized response plans requires contemplation of HCWs’ knowledge and confidence levels to face the potential threats of this re-emerging infection [54]. The frontline position of HCWs requires proper guidance that would help in patient care, in control efforts, and to address the possible issues of burnout and mental health problems frequently encountered among HCWs in outbreak situations [55,56]. Our study results were in agreement with past and recent studies that found defects in knowledge regarding HMPX among the general practitioners in Indonesia, physicians in Italy, and HCWs as well as university students in Jordan [31,33,34,51]. Expectedly, the level of monkeypox knowledge was higher in this study compared to those reported in the general public in Saudi Arabia and in the Kurdistan region of Iraq [30,57].

In this study, gaps in HMPX knowledge were most conspicuous for non-cutaneous manifestations of HMPX, besides the conditions of human-to-human transmission. For the item “human-to-human transmission of monkeypox occurs easily,” less than 40% of the participants responded correctly across the five occupational categories. Similar results were also noticed in the recent studies that used the same knowledge item [34,51]. The relevance of this result is related to the importance of implementing proper control and mitigation measures without exaggeration [58]. Human-to-human transmission of MPXV has been reported prior to the 2022 outbreak, and became evident currently; nevertheless, it should be noted that transmission requires close contact and does not occur as readily as infections caused by respiratory viruses (e.g. SARS-CoV-2) [58,59]. Therefore, the emphasis on providing accurate information about the disease among HCWs cannot be overlooked. Subsequently, this approach among HCWs can help to guide the general public and to provide recommendations for patients, considering their important role amid this outbreak, which should be driven by the accurate knowledge needed to be alert but not panicked [58].

It is also necessary to highlight the importance of providing accurate knowledge and training regarding the clinical presentation and treatment of the disease among HCWs. In this study, a considerable proportion of the participants (31.5%), incorrectly identified diarrhea as a symptom of HMPX. The high index of suspicion is necessary for timely diagnosis of HMPX, with subsequent implementation of control measures including contact tracing and isolation. However, the lack of accurate knowledge regarding the plethora of HMPX clinical manifestations could lead to a waste of valuable resources by the ordering of unnecessary diagnostic tests and the promotion of uneasiness for the patients [60]. An additional important result was the finding that 20.4% of the participants incorrectly identified antibiotics as a treatment for HMPX. This is of particular interest in the Middle East region, where antimicrobial resistance was reported at alarmingly high levels, and the prescription of antibiotics due to lack of knowledge, without being clinically indicated, can aggravate this problem [61,62,63]. 

Despite the general unsatisfactory level of HMPX knowledge observed across different occupational categories in this study, physicians displayed a higher level of knowledge about the disease. This result was in agreement with the findings of a recent study among HCWs in Jordan [51]. The lower level of HMPX knowledge among non-physicians suggests that more efforts are needed to educate and train HCWs in these occupational categories, with a special focus on nurses, considering their central and direct role in patient care and response in outbreak situations [64]. 

In contrast to the pattern of variability in HMPX knowledge per occupation noted in this study, nurses displayed higher self-reported confidence levels to diagnose and manage the disease. This result also contrasts with the findings of the study among HCWs in Jordan, where physicians reported higher confidence levels in association with higher HMPX knowledge, as well [51]. The higher confidence levels among nurses compared to physicians in this study might be ascribed to the large proportion of non-native nurses in Kuwait [65]. Previous studies showed that international experience among nurses is linked with acquisition of new clinical skills and awareness, which could explain such higher confidence levels among nurses [66,67]. The difference between nurses and physicians in this study was less conspicuous for the confidence in diagnostic tests, while generally low levels of confidence were reported among physicians to clinically diagnose and manage HMPX cases based on their current level of knowledge and skills. Thus, focusing on the improvement of physicians’ clinical skills through urgent training workshops and national conferences, as well as on providing clear guidelines for diagnosis and management, are a few suggested intervention measures that could help improve the low levels of confidence observed in this study [32].

An interesting result in this study was the finding that 20.4% of the participants inaccurately believed that HMPX is exclusive to male homosexuals. It was also interesting to note that the belief in this false notion was independently correlated with lower HMPX knowledge and occupation (with higher prevalence of this inaccurate belief among MT/AHP compared to physicians, nurses, and pharmacists). Furthermore, the association between the agreement or neutrality towards the idea of HMPX exclusive occurrence among male homosexuals and the endorsement of conspiracy beliefs regarding emerging virus infections was remarkable, which was also noted in the recent study among HCWs in Jordan [51].

Despite the observation that a majority of HMPX cases amid the current 2022 outbreak involve MSM, the disease was also reported among females and children [12,14,15,68]. Therefore, the idea that HMPX is exclusive to male homosexuals is inaccurate, in spite of the importance of focusing on MSM as the most-at-risk group that should be prioritized for control and preventive efforts, including vaccination [69]. In this study, the survey item that was used to assess knowledge and attitude was not intended to directly assess the attitude towards the MSM role in the outbreak; nevertheless, the results might provide initial clues regarding the view of HCWs towards this issue. Specifically, holding the notion that the current HMPX outbreak is exclusive among MSM can have negative consequences that might include overlooking cases among women and children, in addition to linking the disease with sexual behavior, which might lead to stigma among the infected patients with subsequent deleterious effects on health-seeking behavior [70,71,72]. Thus, future studies are warranted to directly investigate the possible stigma and discrimination directed towards HMPX patients, which is an urgent issue to be addressed, since it can cause collateral psychologic, social, and health damage in addition to the direct impact of the disease [73,74]. Such studies are of particular importance in the majority of countries in the Middle East region as a result of the dominant social, cultural, political, and religious perspectives that oppose homosexuality in the name of religious and cultural integrity [75,76,77]. There is a notable prevalence of stigmatizing attitude towards MSM and patients with sexually transmitted infections in the region [78,79,80]. Subsequently, if MPXV is introduced into the most-at-risk group (i.e. MSM), this can result in an outbreak with exacerbated consequences. Such a worrying outcome could stem from the correlation of stigma with adverse health practices (e.g. unprotected sex, multiple sexual partners, and reduced access to health care services) [81,82,83].

In this study, the embrace of conspiracy beliefs regarding emerging virus infections was independently associated with agreement or lack of opinion towards the role of male homosexuals in HMPX spread. The previous study that was conducted among health professionals in Jordan revealed the same correlation. This result can be interpreted as either false knowledge without further effect on the attitude towards the patients, or as an observation that needs further elaboration to examine its possible link to stigmatizing attitude. Thus, we recommend future studies to examine the effect of such a hypothetical link, considering the importance of this concerning subject. This urgent issue needs to be addressed, taking into account the intensive media focus on MSM’s role in the current outbreak, which can fuel stigma towards this most-at-risk, often highly marginalized group [73].

An important finding of the current study was the wide prevalence adoption of conspiratorial ideas about virus emergence and the subsequent intervention measures, which emerged as a recurrent pattern during the COVID-19 pandemic onwards, and particularly in the Middle East region [34,40,51,84]. Specifically, in this study, the skepticism to the official explanation of virus emergence was found at least to some extent among 60.5% of the study participants. In addition, the belief that “lockdowns in response to emerging infection are aimed for mass surveillance and to control every aspect of our lives” was found among 45.2% of the study sample, besides 44.4% who believed that “viruses are biological weapons manufactured by the superpowers to take global control.” The considerable proportion of individuals who held such beliefs was reported in the original comprehensive study by Freeman et al., with links to medical mistrust and lower levels of compliance to measures aimed to control the COVID-19 pandemic [52].

In this study, and consistent with recent studies among university students and HCWs in Jordan [34,51], greater embrace of these conspiracies about emerging virus infection was associated with lower HMPX knowledge; however, it is important to emphasize that the directionality of this association and the cause–effect relationship cannot be inferred based on the cross sectional study design. Occupational category was also observed as an associated factor with higher embrace of conspiracies among nurses and MT/AHP. Higher embrace of conspiracies was also noted among females. The latter two results were also reported in the context of COVID-19 vaccine hesitancy and vaccine conspiracy beliefs among healthcare workers in Kuwait [85].

It is important to highlight the recurring pattern of misinformation, and sometimes disinformation, that accompanies the reporting of infectious disease emergence [86]. The viral spread of misinformation, amplified through social media, was described as an “infodemic” during the COVID-19 pandemic, and it became evident during the ongoing 2022 multi-country HMPX outbreak [44,45,87,88]. The relevance of this phenomenon in our study was reflected by the substantial percentage of the study participants who endorsed conspiratorial ideas regarding virus origin. The negative impact of such a phenomenon was investigated extensively during the COVID-19 pandemic, which included its association with negative psychologic, social, and health related consequences [42,89,90]. Thus, the finding that a fraction of HCWs—a group that is considered knowledgeable regarding health-related topics—endorsed such ideas, was expected; nevertheless, the high prevalence of adoption of such conspiracies was surprising. One important issue in this context was reported by Alshahrani et al., with social media as the most frequent source of HMPX-related information among the general public in Saudi Arabia [30]. The spread of health-related misinformation, including conspiracies, via social media channels was well known prior to the COVID-19 pandemic and the current HMPX outbreak, and a few suggested responses include the role of health organizations in providing corrective infographics, and partnership with social media platforms to allow immediate action through the promotion of the role of scientific experts in vigilant response [91,92]. 

The study’s strength involved the inclusion of various HCW categories with a large sample size. However, the study was limited by the following caveats that must be considered in the interpretation of results, as follows: (1) Selection bias was inevitable, considering the snowball sampling approach; (2) Despite the large sample size, and the previous evidence that the number of nurses in Kuwait exceed the number of physicians, the dominance of nurses in the current study should be taken into account in the efforts to make generalizations [49]; (3) the cross-sectional design precluded the assessment of causal associations; (4) the social desirability bias can be an important source of bias, particularly in relation to the items assessing conspiracies regarding emerging virus infections; (5) Future knowledge, attitude, and practice (KAP) HMPX studies can benefit from assessment of HMPX vaccination attitudes similar to the approach used by Riccò et al., and the low acceptance of HMPX vaccination in the region, as reported by Sirwan Ahmed et al. in Iraq, [33,57]; (6) The importance of assessment of the stigmatizing attitude towards patients, especially the high-risk MSM groups, should be considered in future research, as well; (7) The possibility of careless responses could not be excluded, particularly in relation to the absence of response time monitoring.

## 5. Conclusions

Knowledge gaps were identified regarding HMPX among a sample of HCWs in Kuwait. This finding needs to be properly addressed with an approach that can be viewed as a trade-off between vigilant and accurate identification of cases without causing panic, while simultaneously avoiding unnecessary exaggerated response. Our findings revealed the low level of confidence regarding HMPX diagnosis and management, especially involving physicians, and this issue needs to be taken into account with proper and timely training and education to facilitate the preparedness of physicians for responding properly to the ongoing outbreak. This poor confidence was associated with a lack of current information and an absence of knowledge regarding how to protect themselves and their patients. A considerable fraction of the study respondents falsely believed that HMPX is exclusive among male homosexuals, and this issue warrants further investigation to assess its potential link with stigma towards the affected patients. The widespread adoption of conspiratorial beliefs regarding emerging virus infections warrants immediate effective interventions, and its impact on the response to HMPX outbreak should be addressed in future studies. 

## Figures and Tables

**Figure 1 pathogens-11-00994-f001:**
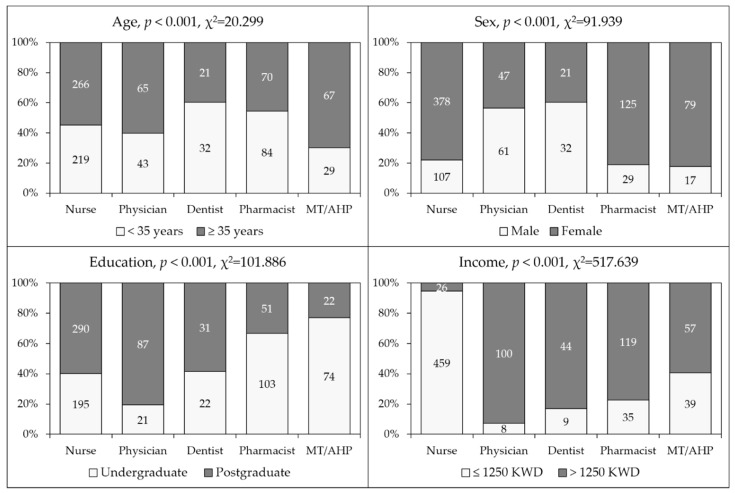
General characteristics of the study sample. MT/AHP: Medical technician/allied health professionals; Education: the highest educational level attained; Income: Monthly income of household; KWD: Kuwaiti dinar.

**Figure 2 pathogens-11-00994-f002:**
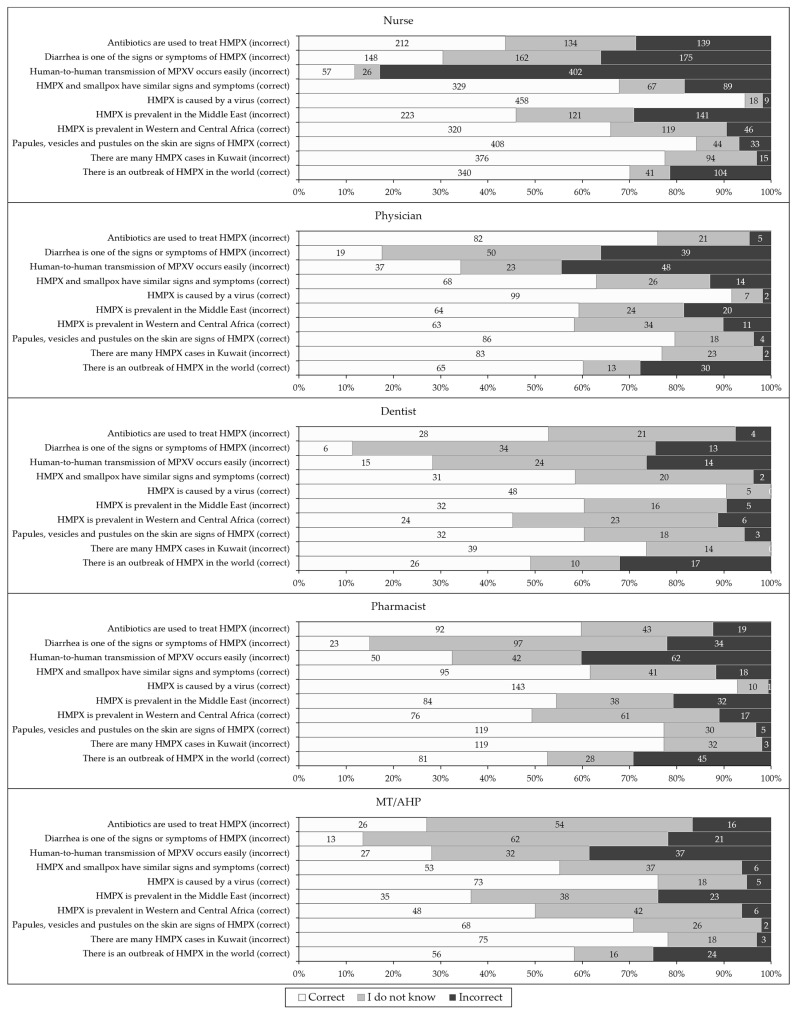
The overall level of human monkeypox (HMPX) knowledge among the study respondents divided by occupational category. MT/AHP: Medical technician/allied health professionals; MPXV: Monkeypox virus.

**Figure 3 pathogens-11-00994-f003:**
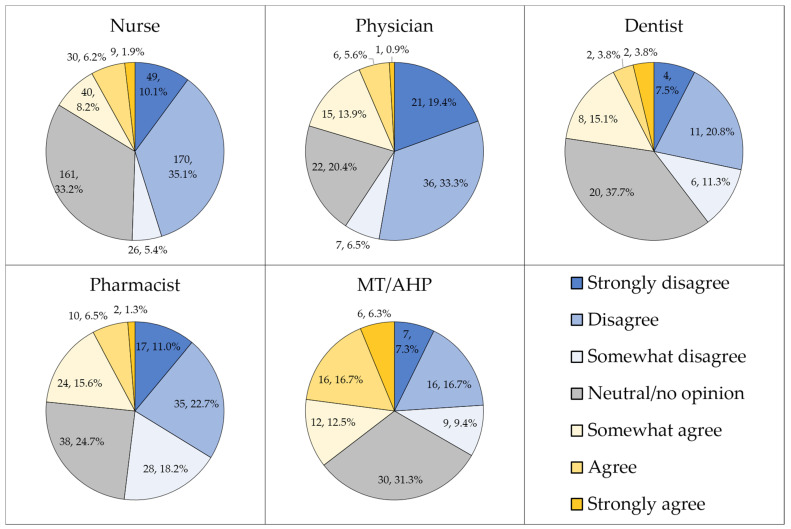
The attitude towards the false notion that human monkeypox (HMPX) is exclusive to male homosexuals. MT/AHP: Medical technician/allied health professionals.

**Figure 4 pathogens-11-00994-f004:**
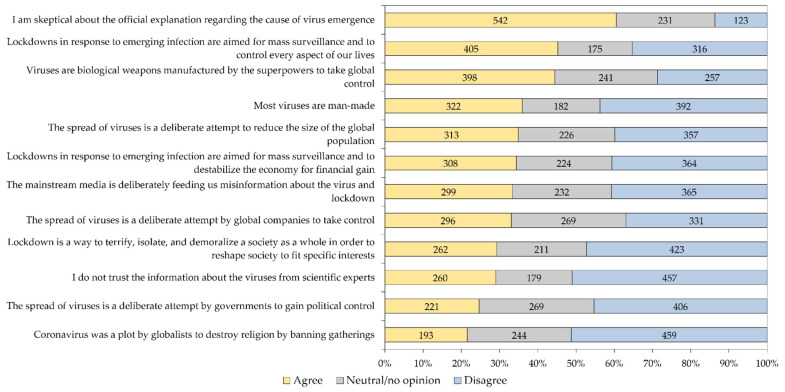
Response of the study participants to the 12 emerging virus infection conspiracy beliefs scale items. Responses were grouped as “Agree” for the three agreement responses (strongly agree, agree and agree to some extent), while the group “Disagree” involved the disagreement responses (strongly disagree, disagree and disagree to some extent).

**Table 1 pathogens-11-00994-t001:** Factors associated with higher human monkeypox (HMPX) knowledge in the whole study sample.

Factors Associated with Higher HMPX K-Score ^1^	Odds Ratio (95% Confidence Interval)	*p* Value
Age < 35 years vs. ≥ 35 years	1.026 (0.782–1.346)	0.852
Males vs. females	1.216 (0.885–1.670)	0.227
Undergraduates vs. postgraduates	0.848 (0.638–1.126)	0.253
Income ≤ 1250 KWD ^2^ vs. > 1250 KWD	1.244 (0.813–1.905)	0.314
Nurses vs. MT/AHP ^3^	0.886 (0.534–1.472)	0.641
Physicians vs. MT/AHP	1.971 (1.056–3.681)	0.033
Dentists vs. MT/AHP	1.443 (0.704–2.957)	0.317
Pharmacists vs. MT/AHP	1.661 (0.979–2.817)	0.060

^1^ The human monkeypox knowledge score (K-score) was divided into two categories: participants with a score <4 (inferior HMPX knowledge) vs. those with a score ≥4 (better HMPX knowledge) with the later as the reference category. ^2^ KWD: Kuwaiti dinar; ^3^ MT/AHP: Medical technicians/allied health professionals.

**Table 2 pathogens-11-00994-t002:** Factors associated with disagreement to the false belief that human monkeypox occurs exclusively among male homosexuals.

Factors Associated with Disagreement Attitude towards the Belief That HMPX Is Exclusive among Male Homosexuals ^1^	Odds Ratio (95% Confidence Interval)	*p* Value
Age < 35 years vs. ≥ 35 years	1.058 (0.807–1.387)	0.683
Males vs. females	0.943 (0.688–1.294)	0.717
Undergraduates vs. postgraduates	1.017 (0.765–1.351)	0.908
Income ≤ 1250 KWD ^2^ vs. > 1250 KWD	0.832 (0.544–1.274)	0.398
Nurses vs. MT/AHP ^3^	2.258 (1.328–3.839)	0.003
Physicians vs. MT/AHP	2.637 (1.403–4.955)	0.003
Dentists vs. MT/AHP	1.221 (0.586–2.543)	0.594
Pharmacists vs. MT/AHP	1.987 (1.155–3.420)	0.013
K-score ^4^ < 4 vs. K-score ≥ 4	0.667 (0.509–0.873)	0.003

^1^ The attitude towards the belief that HMPX is exclusive among male homosexuals was divided into two categories: participants who strongly disagreed, disagreed or somewhat disagreed with this statement vs. those who strongly agreed, agreed or somewhat agreed with this statement or were neutral/had no opinion, with the later as the reference category. ^2^ KWD: Kuwaiti dinar; ^3^ MT/AHP: Medical technicians/allied health professionals; ^4^ K-score: Human monkeypox knowledge score.

**Table 3 pathogens-11-00994-t003:** Responses of the study respondents to the items that assessed confidence to diagnose and manage human monkeypox (HMPX) divided by occupational category.

Confidence Item	Response	Occupational Category					
		Nurse	Physician	Dentist	Pharmacist	MT/AHP ^1^	*p* Value, χ^2^
Are you confident to diagnose monkeypox cases based on your current knowledge and skills?	Yes	222 (45.8)	26 (24.1)	3 (5.7)	21 (13.6)	17 (17.7)	<0.001, 94.752
	No	263 (54.2)	82 (75.9)	50 (94.3)	133 (86.4)	79 (82.3)	
Are you confident to diagnose monkeypox cases based on the ability of your current facility to do diagnostic tests?	Yes	305 (62.9)	55 (50.9)	14 (26.4)	39 (25.3)	36 (37.5)	<0.001, 87.548
	No	180 (37.1)	53 (49.1)	39 (73.6)	115 (74.7)	60 (62.5)	
Are you confident to manage monkeypox cases, if any, based on your current knowledge and skills?	Yes	320 (66.0)	28 (25.9)	9 (17.0)	42 (27.3)	27 (28.1)	<0.001, 146.071
	No	165 (34.0)	80 (74.1)	44 (83.0)	112 (72.7)	69 (71.9)	

^1^ MT/AHP: Medical technicians/allied health professionals.

**Table 4 pathogens-11-00994-t004:** Factors associated with higher confidence to diagnose and manage human monkeypox (HMPX).

Factors Associated with Self-Reported Higher Confidence to Diagnosis and Management of HMPX ^1^	Odds Ratio (95% Confidence Interval)	*p* Value
Age < 35 years vs. ≥ 35 years	0.937 (0.697–1.260)	0.668
Males vs. females	1.212 (0.859–1.711)	0.274
Undergraduates vs. postgraduates	0.619 (0.455–0.843)	0.002
Nurses vs. MT/AHP ^2^	3.603 (2.172–5.975)	<0.001
Physicians vs. MT/AHP	0.749 (0.384–1.463)	0.398
Dentists vs. MT/AHP	0.275 (0.102–0.746)	0.011
Pharmacists vs. MT/AHP	0.563 (0.301–1.056)	0.073
K-score ^3^ < 4 vs. K-score ≥ 4	0.841 (0.626–1.131)	0.252

^1^ The confidence score dichotomized as those with a score of zero or 1 (lower confidence) vs. those with a score of 2 or 3 (higher confidence) with the higher confidence as the reference category; ^2^ MT/AHP: Medical technicians/allied health professionals; ^3^ K-score: Human monkeypox knowledge score.

**Table 5 pathogens-11-00994-t005:** Attitude of the study participants to the 12-item emerging virus infections conspiracy beliefs.

Item	Response	Occupational Category N (%)				
		Nurse	Physician	Dentist	Pharmacist	MT/AHP ^1^
I am skeptical about the official explanation regarding the cause of virus emergence	Strongly disagree	3 (0.6)	6 (5.6)	1 (1.9)	3 (1.9)	2 (2.1)
	Disagree	23 (4.7)	18 (16.7)	5 (9.4)	12 (7.8)	10 (10.4)
	Somewhat disagree	18 (3.7)	7 (6.5)	1 (1.9)	8 (5.2)	6 (6.3)
	Neutral/no opinion	124 (25.6)	30 (27.8)	17 (32.1)	39 (25.3)	21 (21.9)
	Somewhat agree	141 (29.1)	20 (18.5)	17 (32.1)	37 (24.0)	25 (26.0)
	Agree	151 (31.1)	15 (13.9)	7 (13.2)	37 (24.0)	17 (17.7)
	Strongly agree	25 (5.2)	12 (11.1)	5 (9.4)	18 (11.7)	15 (15.6)
I do not trust the information about the viruses from scientific experts	Strongly disagree	39 (8.0)	28 (25.9)	10 (18.9)	23 (14.9)	10 (10.4)
	Disagree	132 (27.2)	38 (35.2)	17 (32.1)	50 (32.5)	12 (12.5)
	Somewhat disagree	46 (9.5)	13 (12)	6 (11.3)	24 (15.6)	9 (9.4)
	Neutral/no opinion	101 (20.8)	12 (11.1)	16 (30.2)	26 (16.9)	24 (25.0)
	Somewhat agree	81 (16.7)	10 (9.3)	2 (3.8)	20 (13)	28 (29.2)
	Agree	78 (16.1)	6 (5.6)	0	6 (3.9)	7 (7.3)
	Strongly agree	8 (1.6)	1 (0.9)	2 (3.8)	5 (3.2)	6 (6.3)
Most viruses are man-made	Strongly disagree	18 (3.7)	32 (29.6)	9 (17.0)	22 (14.3)	8 (8.3)
	Disagree	100 (20.6)	28 (25.9)	20 (37.7)	31 (20.1)	9 (9.4)
	Somewhat disagree	52 (10.7)	18 (16.7)	6 (11.3)	29 (18.8)	10 (10.4)
	Neutral/no opinion	111 (22.9)	10 (9.3)	10 (18.9)	27 (17.5)	24 (25.0)
	Somewhat agree	112 (23.1)	15 (13.9)	6 (11.3)	26 (16.9)	27 (28.1)
	Agree	72 (14.8)	2 (1.9)	1 (1.9)	9 (5.8)	11 (11.5)
	Strongly agree	20 (4.1)	3 (2.8)	1 (1.9)	10 (6.5)	7 (7.3)
The spread of viruses is a deliberate attempt to reduce the size of the global population	Strongly disagree	25 (5.2)	36 (33.3)	13 (24.5)	30 (19.5)	6 (6.3)
	Disagree	75 (15.5)	28 (25.9)	14 (26.4)	36 (23.4)	14 (14.6)
	Somewhat disagree	38 (7.8)	11 (10.2)	3 (5.7)	19 (12.3)	9 (9.4)
	Neutral/no opinion	137 (28.2)	23 (21.3)	15 (28.3)	30 (19.5)	21 (21.9)
	Somewhat agree	103 (21.2)	8 (7.4)	5 (9.4)	18 (11.7)	25 (26)
	Agree	83 (17.1)	1 (0.9)	1 (1.9)	13 (8.4)	11 (11.5)
	Strongly agree	24 (4.9)	1 (0.9)	2 (3.8)	8 (5.2)	10 (10.4)
The spread of viruses is a deliberate attempt by governments to gain political control	Strongly disagree	41 (8.5)	36 (33.3)	12 (22.6)	28 (18.2)	8 (8.3)
	Disagree	127 (26.2)	24 (22.2)	16 (30.2)	32 (20.8)	10 (10.4)
	Somewhat disagree	32 (6.6)	10 (9.3)	2 (3.8)	18 (11.7)	10 (10.4)
	Neutral/no opinion	179 (36.9)	18 (16.7)	13 (24.5)	37 (24)	22 (22.9)
	Somewhat agree	49 (10.1)	15 (13.9)	5 (9.4)	17 (11)	22 (22.9)
	Agree	42 (8.7)	3 (2.8)	4 (7.5)	12 (7.8)	12 (12.5)
	Strongly agree	15 (3.1)	2 (1.9)	1 (1.9)	10 (6.5)	12 (12.5)
The spread of viruses is a deliberate attempt by global companies to take control	Strongly disagree	24 (4.9)	35 (32.4)	9 (17)	25 (16.2)	8 (8.3)
	Disagree	83 (17.1)	22 (20.4)	17 (32.1)	28 (18.2)	11 (11.5)
	Somewhat disagree	28 (5.8)	9 (8.3)	3 (5.7)	20 (13)	9 (9.4)
	Neutral/no opinion	182 (37.5)	25 (23.1)	12 (22.6)	33 (21.4)	17 (17.7)
	Somewhat agree	78 (16.1)	12 (11.1)	9 (17.0)	22 (14.3)	30 (31.3)
	Agree	66 (13.6)	3 (2.8)	1 (1.9)	14 (9.1)	11 (11.5)
	Strongly agree	24 (4.9)	2 (1.9)	2 (3.8)	12 (7.8)	10 (10.4)
Lockdowns in response to emerging infection are aimed for mass surveillance and to control every aspect of our lives	Strongly disagree	12 (2.5)	35 (32.4)	9 (17)	22 (14.3)	9 (9.4)
	Disagree	63 (13.0)	30 (27.8)	15 (28.3)	25 (16.2)	12 (12.5)
	Somewhat disagree	40 (8.2)	6 (5.6)	7 (13.2)	23 (14.9)	8 (8.3)
	Neutral/no opinion	106 (21.9)	13 (12)	10 (18.9)	25 (16.2)	21 (21.9)
	Somewhat agree	105 (21.6)	13 (12)	5 (9.4)	18 (11.7)	21 (21.9)
	Agree	123 (25.4)	8 (7.4)	4 (7.5)	29 (18.8)	16 (16.7)
	Strongly agree	36 (7.4)	3 (2.8)	3 (5.7)	12 (7.8)	9 (9.4)
Lockdowns in response to emerging infection are aimed for mass surveillance and to destabilize the economy for financial gain	Strongly disagree	20 (4.1)	34 (31.5)	8 (15.1)	26 (16.9)	7 (7.3)
	Disagree	89 (18.4)	20 (18.5)	15 (28.3)	32 (20.8)	13 (13.5)
	Somewhat disagree	42 (8.7)	15 (13.9)	12 (22.6)	22 (14.3)	9 (9.4)
	Neutral/no opinion	142 (29.3)	15 (13.9)	12 (22.6)	34 (22.1)	21 (21.9)
	Somewhat agree	95 (19.6)	15 (13.9)	4 (7.5)	19 (12.3)	23 (24.0)
	Agree	83 (17.1)	8 (7.4)	2 (3.8)	14 (9.1)	13 (13.5)
	Strongly agree	14 (2.9)	1 (0.9)	0	7 (4.5)	10 (10.4)
Lockdown is a way to terrify, isolate, and demoralize a society as a whole in order to reshape society to fit specific interests	Strongly disagree	27 (5.6)	37 (34.3)	13 (24.5)	35 (22.7)	8 (8.3)
	Disagree	107 (22.1)	30 (27.8)	13 (24.5)	40 (26.0)	12 (12.5)
	Somewhat disagree	43 (8.9)	13 (12)	7 (13.2)	21 (13.6)	17 (17.7)
	Neutral/no opinion	141 (29.1)	15 (13.9)	17 (32.1)	27 (17.5)	11 (11.5)
	Somewhat agree	91 (18.8)	9 (8.3)	2 (3.8)	14 (9.1)	24 (25)
	Agree	58 (12)	4 (3.7)	0	8 (5.2)	14 (14.6)
	Strongly agree	18 (3.7)	0 (0)	1 (1.9)	9 (5.8)	10 (10.4)
Viruses are biological weapons manufactured by the superpowers to take global control	Strongly disagree	14 (2.9)	23 (21.3)	9 (17.0)	19 (12.3)	6 (6.3)
	Disagree	62 (12.8)	17 (15.7)	10 (18.9)	20 (13)	9 (9.4)
	Somewhat disagree	29 (6.0)	13 (12)	4 (7.5)	18 (11.7)	4 (4.2)
	Neutral/no opinion	143 (29.5)	22 (20.4)	17 (32.1)	32 (20.8)	27 (28.1)
	Somewhat agree	131 (27)	16 (14.8)	9 (17)	29 (18.8)	23 (24)
	Agree	73 (15.1)	13 (12.0)	1 (1.9)	21 (13.6)	13 (13.5)
	Strongly agree	33 (6.8)	4 (3.7)	3 (5.7)	15 (9.7)	14 (14.6)
Coronavirus was a plot by globalists to destroy religion by banning gatherings	Strongly disagree	37 (7.6)	42 (38.9)	21 (39.6)	36 (23.4)	7 (7.3)
	Disagree	133 (27.4)	28 (25.9)	11 (20.8)	43 (27.9)	15 (15.6)
	Somewhat disagree	39 (8.0)	9 (8.3)	6 (11.3)	22 (14.3)	10 (10.4)
	Neutral/no opinion	168 (34.6)	18 (16.7)	9 (17.0)	29 (18.8)	20 (20.8)
	Somewhat agree	55 (11.3)	7 (6.5)	2 (3.8)	7 (4.5)	19 (19.8)
	Agree	37 (7.6)	3 (2.8)	2 (3.8)	9 (5.8)	14 (14.6)
	Strongly agree	16 (3.3)	1 (0.9)	2 (3.8)	8 (5.2)	11 (11.5)
The mainstream media is deliberately feeding us misinformation about the virus and lockdown	Strongly disagree	11 (2.3)	27 (25.0)	7 (13.2)	25 (16.2)	9 (9.4)
	Disagree	90 (18.6)	35 (32.4)	15 (28.3)	38 (24.7)	12 (12.5)
	Somewhat disagree	50 (10.3)	11 (10.2)	6 (11.3)	20 (13.0)	9 (9.4)
	Neutral/no opinion	158 (32.6)	15 (13.9)	12 (22.6)	27 (17.5)	20 (20.8)
	Somewhat agree	112 (23.1)	13 (12.0)	7 (13.2)	24 (15.6)	24 (25)
	Agree	53 (10.9)	4 (3.7)	5 (9.4)	11 (7.1)	15 (15.6)
	Strongly agree	11 (2.3)	3 (2.8)	1 (1.9)	9 (5.8)	7 (7.3)

^1^ MT/AHP: Medical technicians/allied health professionals.

**Table 6 pathogens-11-00994-t006:** Associated factors with embrace of conspiracy beliefs regarding emerging virus infections.

Factors Associated with Higher Embrace of Conspiracy Beliefs about Emerging Virus Infections ^1^	Odds Ratio (95% Confidence Interval)	*p* Value
Age < 35 years vs. ≥ 35 years	0.867 (0.643–1.167)	0.346
Males vs. females	0.697 (0.494–0.984)	0.040
Undergraduates vs. postgraduates	1.270 (0.929–1.738)	0.134
Income ≤ 1250 KWD ^2^ vs. > 1250 KWD	1.246 (0.783–1.984)	0.354
Nurses vs. MT/AHP ^3^	1.019 (0.571–1.819)	0.949
Physicians vs. MT/AHP	0.288 (0.144–0.578)	<0.001
Dentists vs. MT/AHP	0.281 (0.127–0.621)	0.002
Pharmacists vs. MT/AHP	0.420 (0.235–0.749)	0.003
K-score ^4^ < 4 vs. K-score ≥ 4	1.915 (1.424–2.575)	<0.001
Agreement or neutral/no opinion attitude to the exclusivity of HMPX ^5^ occurrence among male homosexuals vs. disagreement with this false notion	3.216 (2.386–4.335)	<0.001

^1^ The emerging virus infections conspiracy scale (EVICS) dichotomized as those with a score of EVICS ≥ 46 indicating a higher embrace of conspiracies vs. those with a score of < 46 indicating a lower embrace of conspiracies with the former as the reference category. ^2^ KWD: Kuwaiti dinar; ^3^ MT/AHP: Medical technicians/allied health professionals; ^4^ K-score: Human monkeypox knowledge score; ^5^ HMPX: Human monkeypox.

## Data Availability

The data presented in this study are available upon request from the corresponding author (M.S.).

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
