# Peer review of "Monkeypox Knowledge and Confidence in Diagnosis and Management with Evaluation of Emerging Virus Infection Conspiracies among Health Professionals in Kuwait"

_pathogens, 2022, doi:10.3390/pathogens11090994_

Round 1

Reviewer 1 Report

In this study, the authors aimed to evaluate human monkeypox (HMPX) knowledge and confidence in diagnosis and management among healthcare workers (HCWs) in Kuwait. This study revealed the presence of knowledge gaps among HCWs in Kuwait regarding the HMPX infection. Therefore, the design of efficient and well-organized response plans requires contemplation of HCWs’ knowledge and confidence levels to face the potential threats of HMPX infection.

 Comments:

Please make sure the results of this study did not overlap with those of ref. 50.

Author Response

Reviewer #1 Comments and Suggestions for Authors

In this study, the authors aimed to evaluate human monkeypox (HMPX) knowledge and confidence in diagnosis and management among healthcare workers (HCWs) in Kuwait. This study revealed the presence of knowledge gaps among HCWs in Kuwait regarding the HMPX infection. Therefore, the design of efficient and well-organized response plans requires contemplation of HCWs’ knowledge and confidence levels to face the potential threats of HMPX infection.

Response: We are deeply thankful for the insightful summary and for the positive critical appraisal of the manuscript by the estimated reviewer.

 Comments:

  1. Please make sure the results of this study did not overlap with those of ref. 50.

Response: Despite using the same survey instrument and the consistent results in terms of evident gaps in HMPX knowledge in the two studies, it was important to assess the levels of monkeypox knowledge and confidence in its diagnosis and management in two different countries with different underlying healthcare systems. One distinct result in the current study compared to the study that was conducted in Jordan was the finding of higher confidence levels (self-reported) to diagnose and manage monkeypox among nurses compared to physicians. This result might point to the recruitment of non-native nurses with higher levels of clinical skills and awareness. Since the reviewer’s comment was relevant, we added the following paragraph and references to the Discussion section (Page 13, lines 354-358): “The higher confidence levels among nurses compared to physicians in this study might be ascribed to the large proportion of non-native nurses in Kuwait [62]. Previous studies showed that international experience among nurses is linked with acquisition of new clinical skills and awareness which could explain such higher confidence levels among nurses [63,64].”

Al-Jarallah, K.F.; Moussa, M.A.A.; Hakeem, S.K.; Al-Khanfar, F.K. The nursing workforce in Kuwait to the year 2020. International Nursing Review 2009, 56,(1): 65-72, doi:10.1111/j.1466-7657.2008.00654.x.

  1. Efendi, F.; Dwi Wahyuni, S.; Indarwati, R.; Hadisuyatmana, S.; Kurniati, A.; Usin, A.Z. The lived experience of Indonesian nurses in Kuwait: A phenomenological study. Kontakt 2020, 22,(4): 235-242, doi:10.32725/kont.2020.040.
  2. Brannan, M. Gaining Perspectives of International Nursing Experiences: a Survey of Registered Nurses. Thesis (Master, Education), Queen's University, Queen's University at Kingston, 2013. Available online: http://hdl.handle.net/1974/7866

Reviewer 2 Report

This study aims to aimed to evaluate monkeypox virus (MPXV) knowledge and confidence in diagnosis and management among healthcare workers (HCWs) in Kuwait.  Authors assert a need for this study due to the rapid rate of disease spread and declaration of the 2022 human monkeypox virus (HMPX) as an international publication health emergency that necessitates a collaborative and well-coordinated response.  Authors further contends that the 2022 HMPX multi-country outbreak has been seen in previously non-endemic countries and territories.  Author state that the few and minimal prior studies conducted among HCWs, students and the general population have shown deficits in HMPX knowledge and confidence.  Authors describe how the rapid increases in cases worldwide require vigilant response plans and preparedness in high income Arab Middle Eastern countries like Kuwait, which can add to the knowledge base.

The review is as follows:

1.     There is a good description of participant recruitment.  What was the response rate for the survey?

2.     For the survey instrument, how many questions were there overall?  How long did it take participants to complete survey?

3.     Lines 166-171 – In “The attitude towards the false belief that HMPX is exclusive among male homosexuals was divided into two categories: participants who strongly disagreed, disagreed or somewhat 1disagreed with the attitude statement (disagreement) vs. those who strongly agreed, agreed or somewhat agreed with the attitude statement or were neutral/had no opinion (agreement/neutral/no opinion)”, would grouping the neutral responses with those that agreed possibly confound the results?

4.     Lines 370-371 – In the HCW “notion that the current HMPX outbreak is exclusive among MSM”, the authors should comment in there are any sociocultural factors within the country that may contribute to this notion.

Overall, this is an insightful, compelling paper on a pertinent topic.  It is well-written and interesting to read.  There are a few clarifying questions to address regarding the methodology and any contextual information that can help give the reader insight into the results interpretation.  This is a solid paper and the authors should be commended for their work.

Author Response

Reviewer #2 Comments and Suggestions for Authors

This study aims to aimed to evaluate monkeypox virus (MPXV) knowledge and confidence in diagnosis and management among healthcare workers (HCWs) in Kuwait.  Authors assert a need for this study due to the rapid rate of disease spread and declaration of the 2022 human monkeypox virus (HMPX) as an international publication health emergency that necessitates a collaborative and well-coordinated response.  Authors further contends that the 2022 HMPX multi-country outbreak has been seen in previously non-endemic countries and territories.  Author state that the few and minimal prior studies conducted among HCWs, students and the general population have shown deficits in HMPX knowledge and confidence.  Authors describe how the rapid increases in cases worldwide require vigilant response plans and preparedness in high income Arab Middle Eastern countries like Kuwait, which can add to the knowledge base.

Response: We are deeply thankful for the positive, comprehensive critical appraisal of the manuscript and for the insightful summary.

The review is as follows:

  1. There is a good description of participant recruitment. What was the response rate for the survey?

Response: The reviewer raised an important point. However, we could not track the number of invitations that were received by potential participants based on the sampling method that relied on snowball sampling and sharing the link in various social media platforms. Thus, we cannot provide an exact percentage that represents the response rate for this survey.

  1. For the survey instrument, how many questions were there overall? How long did it take participants to complete survey?

Response: We would like to thank the reviewer for this comment. Accordingly, we added the following sentence to the Methods section: “The survey comprised five sections, with a total of 32 items including the informed e-consent item…”. Regarding the time needed to complete the survey, the estimated time was about five minutes; however, the Google Forms tool does not feature the time needed to complete the survey to the best of our knowledge. However, we believe that the reviewer raised a relevant and important point in order to monitor potential careless responses. Accordingly, we added this point to the limitations paragraph of the Discussion section (Page 15, lines 463-466): “(7) The possibility of careless responses could not be excluded, particularly in relation to absence of response time monitoring.”

  1. Lines 166-171 – In “The attitude towards the false belief that HMPX is exclusive among male homosexuals was divided into two categories: participants who strongly disagreed, disagreed or somewhat 1disagreed with the attitude statement (disagreement) vs. those who strongly agreed, agreed or somewhat agreed with the attitude statement or were neutral/had no opinion (agreement/neutral/no opinion)”, would grouping the neutral responses with those that agreed possibly confound the results?

Response: We are deeply thankful for this meticulous and important comment. Re-analysis in relation to embrace of conspiracy beliefs regarding emerging virus infections did not yield different results than those initially presented in the manuscript except for the slight changes in OR, 95%CI and the p values. Please see (Table 1) below that represents the repeated analysis based on the reviewer suggestion and please compare the results to those present in (Table 6) of the revised highlighted manuscript (11). Accordingly, we prefer to keep the analysis in the current form since the main concept that adoption of such false belief was associated with higher prevalence of conspiratorial ideas regarding virus emergence.

Table 1. Associated factors with embrace of conspiracy beliefs regarding emerging virus infections.

Factors associated with higher embrace of conspiracy beliefs about emerging virus infections 1

Odds ratio (95% confidence interval)

p value

Age < 35 years vs. ≥ 35 years

0.868 (0.650–1.158)

0.335

Males vs. females

0.712 (0.509–0.994)

0.046

Undergraduates vs. postgraduates

1.269 (0.938–1.718)

0.122

Income ≤ 1250 KWD 2 vs. > 1250 KWD

1.299 (0.830–2.034)

0.253

Nurses vs. MT/AHP 3

0.888 (0.506–1.560)

0.680

Physicians vs. MT/AHP

0.262 (0.134–0.514)

<0.001

Dentists vs. MT/AHP

0.307 (0.142–0.662)

0.003

Pharmacists vs. MT/AHP

0.386 (0.220–0.679)

0.001

K-score 4 < 4 vs. K-score ≥ 4

2.067 (1.551–2.755)

<0.001

Agreement attitude to the exclusivity of HMPX 5 occurrence among male homosexuals vs. disagreement/ or having neutral/no opinion with this false notion

1.683 (1.167–2.429)

0.005

1 The emerging virus infections conspiracy scale (EVICS) dichotomized as those with a score of EVICS ≥ 46 indicating a higher embrace of conspiracies vs. those with a score of < 46 indicating a lower embrace of conspiracies with the former as the reference category. 2 KWD: Kuwaiti dinar; 3 MT/AHP: Medical technicians/allied health professionals; 4 K-score: Human monkeypox knowledge score; 5 HMPX: Human monkeypox.

  1. Lines 370-371 – In the HCW “notion that the current HMPX outbreak is exclusive among MSM”, the authors should comment in there are any sociocultural factors within the country that may contribute to this notion.

Response: We would like to thank the reviewer for raising this important point that indeed was worth of further elaboration considering the study setting in an Arab, Middle Eastern country. Accordingly, we added the following paragraph and references to the Discussion section (Page 13, lines 390-398): “Such studies are of particular importance in the majority of countries in the Middle East region as a result of dominant social, cultural, political and religious perspectives with opposition to homosexuality paralleling religious and cultural integrity [73-75]. There is a notable prevalence of stigmatizing attitude towards MSM and patients with sexually transmitted infections in the region [76-78]. Subsequently, if MPXV is introduced into the most-at-risk group (i.e. MSM), this can result in an outbreak with exacerbated consequences. Such a worrying outcome could stem from the correlation of stigma with adverse health practices (e.g. unprotected sex, multiple sexual partners and reduced access to health care services) [79-81].”

  1. Dalacoura, K. Homosexuality as cultural battleground in the Middle East: culture and postcolonial international theory. Third World Quarterly 2014, 35,(7): 1290-1306, doi:10.1080/01436597.2014.926119.
  2. Pratt, N. The Queen Boat case in Egypt: sexuality, national security and state sovereignty. Review of International Studies 2007, 33,(1): 129–144, doi:10.1017/S0260210507007346.
  3. Kligerman, N. Homosexuality in Islam: A Difficult Paradox. Macalester Islam Journal 2007, 2,(3): 8, doi:Available at: https://digitalcommons.macalester.edu/islam/vol2/iss3/8.
  4. Abu-Raddad, L.J.; Ghanem, K.G.; Feizzadeh, A.; Setayesh, H.; Calleja, J.M.G.; Riedner, G. HIV and other sexually transmitted infection research in the Middle East and North Africa: promising progress? Sex Transm Infect 2013, 89,(Suppl 3): iii1, doi:10.1136/sextrans-2013-051373.
  5. Sallam, M.; Alabbadi, A.M.; Abdel-Razeq, S.; Battah, K.; Malkawi, L.; Al-Abbadi, M.A.; Mahafzah, A. HIV Knowledge and Stigmatizing Attitude towards People Living with HIV/AIDS among Medical Students in Jordan. International Journal of Environmental Research and Public Health 2022, 19,(2): 745, doi:10.3390/ijerph19020745.
  6. Wagner, G.J.; Aunon, F.M.; Kaplan, R.L.; Karam, R.; Khouri, D.; Tohme, J.; Mokhbat, J. Sexual stigma, psychological well-being and social engagement among men who have sex with men in Beirut, Lebanon. Cult Health Sex 2013, 15,(5): 570-582, doi:10.1080/13691058.2013.775345.
  7. El-Sayyed, N.; Kabbash, I.A.; El-Gueniedy, M. Risk behaviours for HIV/AIDS infection among men who have sex with men in Cairo, Egypt. East Mediterr Health J 2008, 14,(4): 905-915, doi:n/a.
  8. Caballero-Hoyos, R.; Monárrez-Espino, J.; Ramírez-Ortíz, M.G.; Cárdenas-Medina, F.M. Factors Associated with Unprotected Anal Sex among Men Who Have Sex with Men in Mexico. Infectious Disease Reports 2022, 14,(4): 547-557, doi:10.3390/idr14040058.
  9. Pachankis, J.E.; Hatzenbuehler, M.L.; Hickson, F.; Weatherburn, P.; Berg, R.C.; Marcus, U.; Schmidt, A.J. Hidden from health: structural stigma, sexual orientation concealment, and HIV across 38 countries in the European MSM Internet Survey. AIDS 2015, 29,(10): 1239-1246, doi:10.1097/qad.0000000000000724.

Overall, this is an insightful, compelling paper on a pertinent topic.  It is well-written and interesting to read.  There are a few clarifying questions to address regarding the methodology and any contextual information that can help give the reader insight into the results interpretation.  This is a solid paper and the authors should be commended for their work.

Response: Thanks a lot for your insightful and constructive comments. We are deeply grateful for the time dedicated to review our manuscript.

Reviewer 3 Report

The manuscript proposes an interesting study on the evaluation of the knowledge about the human monkeypox (HMPX) disease in diagnosis and management among workers in the healthcare system in Kuwait.

The manuscript is well written and organized, fitting well with the journal’s topics.

Some minor concerns are related to the following aspects:

·        The abstract appears too long: it should be condensed following the journal’s editorial rules.

·        In the introduction, the research motivations should be improved, bringing to light the novelty of the current study compared to the existing literature.

·        The numbering of sections and subsections needs a revision since different mistakes can be found in the text (e.g. lines 159, 190, 219, 237).

·        Additional information about the questionnaire should be provided in the text to better understand both the value of the analysis and the results achieved.

Author Response

Reviewer #3 Comments and Suggestions for Authors

The manuscript proposes an interesting study on the evaluation of the knowledge about the human monkeypox (HMPX) disease in diagnosis and management among workers in the healthcare system in Kuwait.

The manuscript is well written and organized, fitting well with the journal’s topics.

Response: We are deeply thankful for the summary of the manuscript.

Some minor concerns are related to the following aspects:

  1. The abstract appears too long: it should be condensed following the journal’s editorial rules.

Response: Thanks for this comment, and accordingly, we tried to work on the abstract in order to condense its content without compromising the major results and conclusions. Please check the revised highlighted manuscript (Page 1).

  1. In the introduction, the research motivations should be improved, bringing to light the novelty of the current study compared to the existing literature.

Response: We are thankful for this comment, and we agree with the reviewer. Based on that, we added the following sentences to the Introduction section (Page 2, lines 88-90): “Further research is needed, particularly among health professionals based on their central role in outbreak response and due to the rapid dissemination of the HMPX outbreak [36].” And (Page 3, lines 99-101): “Therefore, the assessment of conspiracy beliefs pervasiveness, especially among HCWs appears essential considering their role to curtail the negative influence of these harmful beliefs [46].”

  1. Ogoina, D.; Izibewule, J.H.; Ogunleye, A.; Ederiane, E.; Anebonam, U.; Neni, A.; Oyeyemi, A.; Etebu, E.N.; Ihekweazu, C. The 2017 human monkeypox outbreak in Nigeria-Report of outbreak experience and response in the Niger Delta University Teaching Hospital, Bayelsa State, Nigeria. PLOS One 2019, 14,(4): e0214229, doi:10.1371/journal.pone.0214229.
  2. Leonard, M.J.; Philippe, F.L. Conspiracy Theories: A Public Health Concern and How to Address It. Front Psychol 2021, 12,(n/a): 682931, doi:10.3389/fpsyg.2021.682931.
  3. The numbering of sections and subsections needs a revision since different mistakes can be found in the text (e.g. lines 159, 190, 219, 237).

Response: We apologize for this mistake and we are thankful for the reviewer who pointed to these mistakes. Accordingly, we corrected the section and sub-section numbering.

  1. Additional information about the questionnaire should be provided in the text to better understand both the value of the analysis and the results achieved.

Response: We are thankful for this comment; however, we respectfully disagree with the reviewer since the Methods section provide a detailed description of the study questionnaire in terms of items used, sampling approach, eligibility criteria and the primary study measures. Please check the following paragraphs in the revised highlighted manuscript: Section 2.1. Study Design; Section 2.2. Survey Instrument; and 2.3. Statistical Analysis.

Thanks again for the positive and constructive feedback on the manuscript.